# Ortholog Analysis and Transformation of Glycoside Hydrolase Genes in Hyperthermophilic Archaeal *Thermococcus* Species

**DOI:** 10.3390/ijms26073305

**Published:** 2025-04-02

**Authors:** Jun Won Lee, Jae Kyu Lim, Hyun Sook Lee, Sung Gyun Kang, Jung-Hyun Lee, Kae Kyoung Kwon, Yun Jae Kim

**Affiliations:** 1Marine Biotechnology Research Center, Korea Institute of Ocean Science and Technology, Busan 49111, Republic of Korea; ljunwon1114@kiost.ac.kr (J.W.L.); leeh522@kiost.ac.kr (H.S.L.); sgkang@kiost.ac.kr (S.G.K.); jlee@kiost.ac.kr (J.-H.L.); kkkwon@kiost.ac.kr (K.K.K.); 2Jeju Bio Research Center, Korea Institute of Ocean Science and Technology, Jeju 63349, Republic of Korea; j.k.lim@kiost.ac.kr; 3KIOST School, University of Science and Technology, Daejeon 34113, Republic of Korea

**Keywords:** *Thermococcus*, ortholog analysis, GH gene family, hyperthermophilic CAZymes, hyperthermophilic archaea

## Abstract

Archaea thrive in extreme environments, exhibiting unique traits with significant biotechnological potential. In this study, we investigated whether *Thermococcus onnurineus* NA1 could stably integrate a large glycoside hydrolase (GH) gene cluster from *T. pacificus* P-4, enhancing β-linked polysaccharides degradation for hydrogen production. Among 35 *Thermococcus* genomes examined via OrthoFinder2 and OrthoVenn3, and selecting *Tpa*-GH gene clusters as the target, we cloned and integrated *Tpa*-GH into *T. onnurineus* NA1 using a fosmid-based system, creating the GH03 mutant. Cultivation in a modified MM1 medium supplemented with laminarin revealed significantly higher growth and hydrogen production in *T. onnurineus* GH03 than in the wild-type strain. Our findings demonstrate the feasibility of stable, large-fragment DNA integration in hyperthermophilic archaea and underscore the promise of *T. onnurineus* GH03 as a strain for high-temperature biomass conversion.

## 1. Introduction

Archaea comprise a domain of single-celled microorganisms that are prokaryotic and possess unique biochemical and physiological properties similarly to bacteria. These characteristics allow archaea to thrive in some of the most extreme environments on Earth [1,2]. Due to these adaptations to various environments, distinctive features have evolved in archaea, which are valuable resources for exploring new biotechnological applications [3]. Recent studies have revealed the roles of archaea in environmental bioremediation, particularly in the breakdown of hydrocarbons, metal detoxification, dehalogenation, and the treatment of acid mine drainage [4,5,6,7,8]. In marine environments, archaea are found in diverse habitats ranging from polar to tropical regions and can constitute up to 30% of the total prokaryotic rRNA in these ecosystems [9,10]. They play roles in biogeochemical processes, particularly in energy cycling such as nitrogen and carbon cycling [11,12].

Archaea isolated from hyperthermal environments such as hot springs or volcanic areas possess hyperthermophilic traits, allowing their proteins to function effectively in high-temperature industrial processes [13]. Whereas thermophiles generally grow optimally between 45 °C and 70 °C, these hyperthermophiles exhibit an even greater thermal tolerance and typically thrive in extremely hot environments, above 80 °C [14,15,16,17]. Members of the genus *Thermococcus* are known for their ability to survive and function in extreme heat. They are often isolated from hyperthermal environments such as deep-sea and terrestrial hot vents [18]. Their unique metabolic properties allow them to endure and adapt to extreme conditions, contributing to their ecological role in such environments [19]. Research has shown that *Thermococcus* species can ferment various carbohydrates, including glucose, starch, and xylans, by utilizing a range of carbohydrate-active enzymes (CAZymes) [20,21].

CAZymes derived from hyperthermophilic archaea exhibit enhanced thermal stability compared with those from other organisms [22]. Sequence-based analyses have explored the biodiversity of hydrothermal vents by characterizing novel hyperthermophilic CAZymes [23]. These enzymes facilitate heterotrophic growth at elevated temperatures, demonstrating their ability to thrive in extreme conditions [22,24]. Ongoing research in this area is crucial, as these efforts aim to harness their industrial and biotechnological potential [20,25].

Glycoside hydrolases, also known as glycosidases or glycosyl hydrolases, are present in most living organisms and play vital roles in metabolic processes, antimicrobial defense, and pathogenic mechanisms [26,27,28,29,30]. β-glucosidases, specifically, hydrolyze β-1,4 glycosidic bonds in substrates such as cellulose or laminarin, releasing glucose monomers. However, the broader utility of many β-glucosidases is restricted due to their low thermostability and glucose tolerance, particularly in high-temperature and high-glucose environments [31]. These enzymes have been identified and characterized in *Thermococcus* species [32,33], in which they exhibit unique features such as a relatively low Km value, stability at extremely high temperatures, and tolerance to elevated glucose concentrations [33,34].

*Thermococcus onnurineus* NA1 has been isolated from a deep-sea hydrothermal vent in the PACMANUS hydrothermal field [35]. This strain is known for its ability to thrive in high-temperature anaerobic environments and has attracted interest due to its versatile metabolism; in particular, its production of hydrogen gas as a metabolic byproduct [36,37,38] holds significant potential for bioenergy applications [39]. Due to its resilience and metabolic adaptability, provided by its various enzymatic systems that enable it to utilize substrates under extreme conditions, *T. onnurineus* NA1 is an excellent model organism for studies exploring genetic transformations to enhance the breakdown of biomass, including complex carbohydrates, and to maximize sustainable hydrogen production [40,41].

In this study, we pursued two primary objectives. The first was to determine whether *T. onnurineus* NA1 could stably integrate and express a large gene cluster, thus serving as a hyperthermophilic strain for potential applications for the conversion of biomass to hydrogen. The second was to enhance the glycoside hydrolase degradation capabilities of *T. onnurineus* NA1 by introducing glycoside hydrolase genes from *T. pacificus* P-4. Therefore, we collected the genomes of various *Thermococcus* species and identified orthologs of glycoside hydrolase genes. We selected candidate genes from *T. pacificus* P-4 and transformed them into wild-type *T. onnurineus*. We then investigated the impact of the introduced glycoside hydrolase genes on the degradation capacity of the resulting mutant, *T. onnurineus* GH03.

## 2. Results

### 2.1. Ortholog Detection in Thermococcus Species

From the genomes of 35 species of *Thermococcus*, OrthoFinder assigned 74,359 genes to 4924 orthogroups, and 3741 orthogroups were assigned as cluster groups (76.0% of the total). More than 50 percent of the total orthogroups (*n* = 2673) included 6 or more genes, and the largest orthogroup contained 181 gene clusters, including duplicated genes. A total of 742 orthogroups consisted of single-copy genes. The accession numbers and details of the 35 *Thermococcus* species used in this analysis are provided in Table 1.

The *Thermococcus* species, including those with orthogroups containing the β-glucosyl hydrolase gene, were identified using InterProScan and NCBI BLAST (Table 2). Seven β-glucosyl hydrolase-related orthogroups were selected, and *T. pacificus* P-4 was found to contain the largest number of β-glucosyl hydrolase-related genes (Table 1). One of these clusters encompassed the thermally stable β-glucosyl hydrolase gene (marked with an unknown GH domain) previously identified by our research group in *Thermococcus pacificus* P-4 [34]. We also confirmed that *T. onnurineus* NA1 does not carry these genes. We conducted pairwise comparisons to explore genomic differences between the two strains.

The pairwise comparisons between *T. pacificus* P-4 and *T. onnurineus* NA1 revealed 1480 shared orthogroup gene clusters. Additionally, *T. pacificus* P-4 possessed 20 unique orthogroups, 2 of which were identified as containing β-glucoside hydrolase-associated genes, while *T. onnurineus* NA1 had 16 unique orthogroups (Figure 1). These gene clusters were selected and named *Tpa*-GH gene clusters.

### 2.2. Incorporation of the Tpa-GH Gene Cluster in the Genome of T. onnurineus NA1

The hyperthermophilic archaeon *Thermococcus pacificus* P-4 possessed an approximately 20 kbp glycosyl hydrolase (*Tpa*-GH) gene cluster, which included 7 genes responsible for the degradation of cellulose, laminarin, and agarose [34] (Figure 2A). The fosmid vector was used as a large-capacity gene-cloning vector for constructing mutants of *T. onnurineus* NA1 in a previous study [42]. In this study, the modified fosmid vector pNA1ComfosC1096 was also utilized to facilitate the cloning and chromosomal insertion of the GH gene cluster from *T. pacificus* into the genome of *T. onnurineus* NA1. The 20 kbp *Tpa*-GH gene cluster was amplified as 2 PCR products, each approximately 10 kbp in size: *Tpa*-GH region I (PAC_orf01349~PAC_orf01356) and *Tpa*-GH region II (PAC_orf01357~PAC_orf01364). First, the PCR-amplified, 10,049 bp *Tpa*-GH region I was cloned into pNA1ComfosC1096, constructing pTpaGH1. The *Mlu*I restriction enzyme site was incorporated at the 3′ end of *Tpa*-GH region I using a reverse PCR primer to enable fosmid cleavage for the subsequent cloning of *Tpa*-GH region II. In the second cloning step, the 10,648 bp *Tpa*-GH region II, amplified by PCR, was cloned directly downstream of *Tpa*-GH region I in the linearized pTpaGH1 vector, following digestion with the *Mlu*I restriction enzyme. This resulted in the construction of the pPAC-GHC1115 fosmid, which contained the complete 20 kbp *Tpa*-GH gene cluster and the HMG cassette, conferring resistance to simvastatin as a selectable marker (Figure 2B). The *Tpa*-GH gene cluster was then integrated into the chromosomal region between the convergent genes TON_1126 and TON_1127 in *T. onnurineus* NA1 via homologous recombination, using pPAC-GHC1115 as the transformation vector and following the strategy used in [37]. As a result, the mutant strain *T. onnurineus* GH03, containing the *Tpa*-GH gene cluster integrated into its genome, was successfully constructed (Figure 3).

**Table 1 ijms-26-03305-t001:** Accession numbers of the Thermococcus species used in this study, along with the number of genes each species contains in the corresponding orthogroups.

Species	Strain	Accession	OG0000069	OG0001332	OG0002060	OG0002308	OG0002385	OG0002611	OG0003709	Total Genes
*T. aciditolerans*	SY113	GCF_008152015	1	1	1	0	1	0	1	5
*T. aggregans*	TY	GCF_024022995	1	0	0	0	0	0	0	1
*T. alcaliphilus*	AEDII12	GCF_024054535	1	0	0	0	0	0	0	1
*T. argininiproducens*	IOH2	GCF_023746595	1	0	0	0	0	0	1	2
*T. barophilus*	MP	GCF_000151105	1	1	0	0	1	0	0	3
*T. barossii*	SHCK-94	GCF_002214465	2	1	0	0	1	0	0	4
*T. bergensis*	T7324	GCF_020386975	1	0	0	0	0	0	0	1
*T. camini*	IRI35c	GCF_904067545	1	1	1	0	1	0	0	4
*T. celer*	Vu13	GCF_002214365	2	1	0	1	0	1	0	5
*T. celericrescens*	DSM17994	GCF_001484195	2	1	1	0	1	0	0	5
*T. chitonophagus*	GC74	GCF_002214605	2	1	1	0	0	0	0	4
*T. cleftensis*	CL1	GCF_000265525	0	1	1	0	0	0	0	2
*T. eurythermalis*	A501	GCF_000769655	1	1	1	0	0	0	0	3
*T. gammatolerans*	EJ3	GCF_000022365	0	1	0	0	0	0	0	1
*T. gorgonarius*	W-12	GCF_002214385	2	1	0	1	0	1	0	5
*T. guaymasensis*	DSM11113	GCF_000816105	2	1	0	1	0	1	0	5
*T. henrietii*	EXT12c	GCF_900198835	1	1	1	0	0	0	0	3
*T. indicus*	IOH1	GCF_006274605	1	1	1	0	0	0	0	3
*T. kodakarensis*	KOD1	GCF_000009965	2	1	1	0	0	0	0	4
*T. litoralis*	DSM 5473	GCF_000246985	2	0	0	0	0	0	0	2
*T. nautili*	01-30	GCF_000585495	1	1	1	0	0	0	0	3
*T. onnurineus*	NA1	GCF_000018365	0	2	0	0	0	0	0	2
*T. pacificus*	P-4	GCF_002214485	4	1	1	2	0	1	0	9
*T. paralvinellae*	ES1	GCF_000517445	0	1	0	0	0	0	0	1
*T. peptonophilus*	OG-1	GCF_001592435	0	1	0	0	0	0	0	1
*T. piezophilus*	CDGS	GCF_001647085	0	1	0	0	0	0	0	1
*T. profundus*	DT5432	GCF_002214585	2	1	0	0	0	0	0	3
*T. radiotolerans*	EJ2	GCF_002214565	2	1	0	0	0	0	0	3
*T. sibiricus*	MM739	GCF_000022545	3	0	0	3	0	1	0	7
*T. siculi*	RG-20	GCF_002214505	1	1	1	0	1	0	0	4
*T. stetteri*	DSM5262	GCF_017873335	2	2	1	0	0	0	0	5
*T. thermotolerans*	813A4	GCF_024707485	1	1	0	0	1	0	0	3
*T. thioreducens*	OGL-20P	GCF_900109425	1	1	0	0	1	0	0	3
*T. waiotapuensis*	WT1	GCF_032304395	2	1	0	1	0	1	0	5
*T. zilligii*	AN1	GCF_000258515	1	1	0	0	0	0	0	2

**Figure 1 ijms-26-03305-f001:**
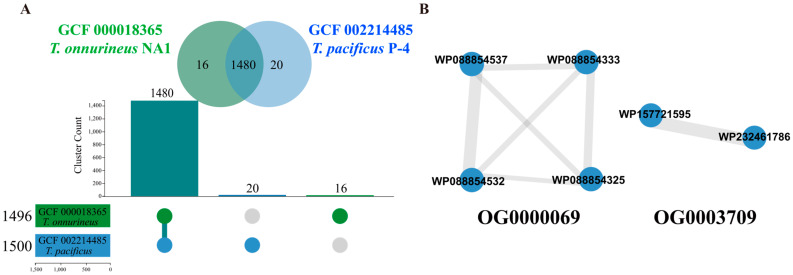
The results of ortholog analysis. (**A**) The comparison results of orthologous analysis between *T. pacificus* P-4 and *T. onnurineus* NA1. (**B**) GH gene clusters of the OG0000069 and OG0003709 orthogroups from *T. pacificus* P-4.

**Table 2 ijms-26-03305-t002:** The number of species that contain β-glucosyl hydrolase gene orthogroups among 35 *Thermococcus* species.

Orthogroup (Gene Cluster)	OG0000069	OG0001332	OG0002060	OG0002385	OG0002611	OG0003709	OG0002308
Gene family	GH1 family	CE4 family	GH38, GH57 family	GH4 family	GH16 family	GH5, GH12 family	Unknown GH domain
Number of species (spp.)	29	29	13	8	6	2	6

**Figure 2 ijms-26-03305-f002:**
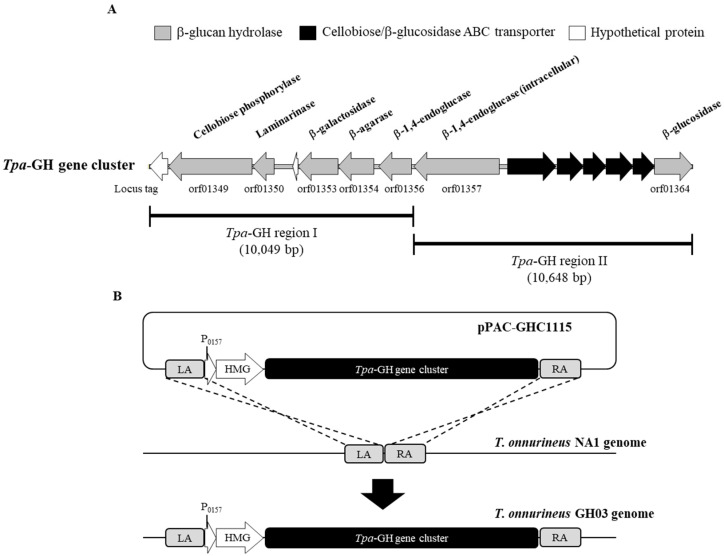
Schematic diagram of mutant construction. (**A**) Diagram of the β-glycosyl hydrolase gene cluster in *T. pacificus*. PCR-amplified *Tpa*-GH region I and *Tpa*-GH region II are indicated below the gene cluster. (**B**) Construction of *T. onnurineus* GH03 mutants by transformation and homologous recombination of the *Tpa*-GH gene cluster containing fosmids pPAC-GHC1115.

**Figure 3 ijms-26-03305-f003:**
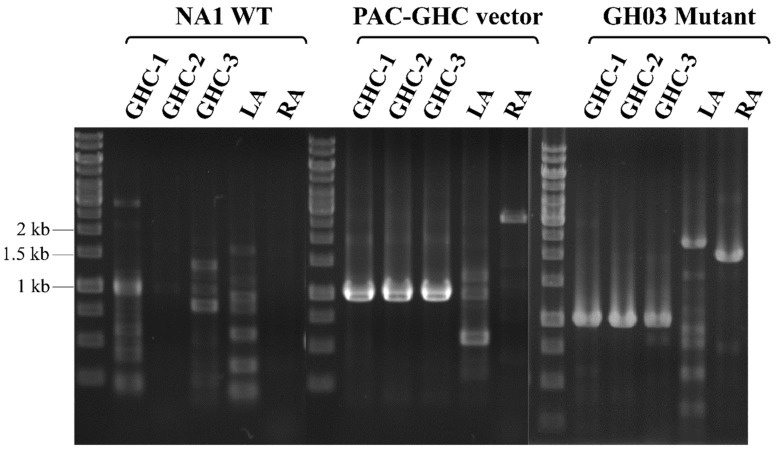
Construction of *T. onnurineus* GH03 mutant. Agarose gel verification of the cloned *Tpa*-GH gene cluster vector and its integration into *T. onnurineus* NA1. Detailed information on the fosmid vector and PCR primers is provided in Appendix A, respectively.

### 2.3. Enhanced Growth and Hydrogen Generation in the T. onnurineus GH03 Mutant

We compared the growth (OD_600_) and hydrogen production of the wild-type *T. onnurineus* and GH03 mutant strains in MM1 media supplemented with carboxymethylcellulose, laminarin, or agarose (Figure 4). Overall, the GH03 mutant exhibited markedly enhanced growth and hydrogen production, especially in the presence of laminarin. Under laminarin-supplemented conditions (MM1-Lam), the mutant’s OD_600_ reached approximately 0.06, whereas the wild-type *T. onnurineus* showed only a minimal increase. Consistently, the hydrogen yields for the GH03 mutant in MM1-Lam approached more than 3 mmol/L.

## 3. Discussion

Ortholog analysis was conducted for the *Thermococcus* genome, and the GH gene cluster from *T. pacificus* (*Tpa*-GH gene cluster) was successfully transformed into the *T. onnurineus* GH03 mutant. As a result, both the growth of the *T. onnurineus* GH03 mutant and its H_2_ production were higher in the laminarin-based medium compared with those of the wild-type *T. onnurineus* NA1. These results indicate that *T. onnurineus* GH03 carrying the introduced glycoside hydrolase genes could more effectively utilize laminarin for growth and hydrogen generation than its wild type. We suggest that introducing additional genes through transformation may enable the *T. onnurineus* NA1 strain to degrade laminarin and enable a broader range of glucosidic bonds in various biomass sources.

The ability of *Thermococcus* species to efficiently degrade laminarin demonstrates their metabolic adaptability to marine environments [34]. Laminarin, a polysaccharide primarily found in brown algae, has a backbone composed mainly of β-1,3-glucose linkages and partially of β-1,6-glucose linkages [43,44]. Previous studies have revealed that several enzymes do not hydrolyze polysaccharides and show minimal activity toward β-1,3-glucans such as laminarin [45,46], and the β-glucosidase from the mesophilic bacterium *Cellvibrio mixtus* has been reported to effectively hydrolyze laminarin as a substrate in only one study [47]. The presence of laminarin-degrading enzymes in *Thermococcus* species suggests that these organisms play an essential role in the carbon cycle within hydrothermal vent ecosystems [48]. In these environments, marine polysaccharides from algal laminarin could serve as accessible carbon and energy sources [49]. This capability suggests that *Thermococcus* species may contribute to the breakdown of complex polysaccharides in high-temperature, nutrient-limited environments and could have potential applications in biotechnology. Given the structural complexity of laminarin, its efficient hydrolysis by *Thermococcus onnurineus* GH03 enzymes may provide insights into novel, thermostable glycoside hydrolases that could be valuable in industrial processes requiring high-temperature biocatalysts. In addition, the monosaccharides obtained from breaking down laminarin can produce biofuels with hydrogen as a byproduct [50].

The *Tpa*-GH gene cluster introduced into *T. onnurineus* GH03 displays high primary-sequence similarity to *Pyrococcus furiosus* β-glucosidases, yet it confers a markedly different substrate specificity [51]. Previous work suggests this discrepancy may stem from structural variations in a loop region at the *A*–*C* dimer interface in the β-glucosidase tetramer, where multiple ion-pair interactions occur [34]. This loop is hypothesized to interact with the fourth barrel helix in *Tpa*-GH, forming an extensive ion-pair network that modifies the binding or catalysis of β-1,3-glucans. Consistent with these structural insights, the *T. onnurineus* GH03 mutant exhibits superior laminarin hydrolysis relative to the wild type. Further structural or mutational analyses are needed to clarify the substrate range of the *Tpa*-GH gene and enable strategic enzyme engineering for broader polysaccharide applications.

The successful transformation of the large *Tpa*-GH gene cluster (>20 kbp) between *Thermococcus* species represents a potential advancement in genetic engineering for archaeal strains and shows that these hyperthermophilic archaea can accept and maintain large DNA inserts. While this procedure is usually challenging due to difficulties in achieving the stable integration and expression of long gene sequences, the realization of these processes in large gene clusters in *Thermococcus onnurineus* NA1 suggests its potential as a genetic tool for developing a hyperthermophilic strain capable of hydrogen production from various biomasses [37,38]. These findings provide a basis for advancing the synthetic biology and biotechnological applications of *Thermococcus* species in extreme environments.

Beyond enhancing laminarin degradation and hydrogen production, successfully integrating a > 20 kbp gene cluster in *T. onnurineus* NA1 opens possibilities for large-scale metabolic engineering in hyperthermophiles. Further applications may include the breakdown of diverse polysaccharides such as cellulose, xylan, and agarose, leveraging the robust thermostability of archaeal enzymes. This expanded substrate range, coupled with enhanced H_2_ production, underscores the potential of *Thermococcus* as a bioresource for producing environmentally friendly hydrogen fuel from diverse renewable biomasses [37,38]. This study shows the potential of *Thermococcus* species as versatile platforms for sustainable energy generation, highlighting the need for the continued development of genetic tools and bioprocess strategies suited to high-temperature environments.

*T. onnurineus* NA1 is widely recognized for its hydrogen production using formate as a substrate; however, it has previously shown a limited ability to utilize β-linked polysaccharides. We focused on expanding the strain’s substrate range by engineering glycoside hydrolase activity. As a result, *T. onnurineus* GH03 demonstrated improved laminarin degradation compared with its wild type, although this enhancement was coupled with reduced hydrogen production [52,53]. Moreover, its laminarin-degrading efficiency remained lower than strains specifically optimized for laminarin utilization [54], suggesting that further adaptive evolution or metabolic engineering could improve laminarin hydrolysis. Although *Tpa*-GH exhibits high primary-sequence similarity to *Pyrococcus furiosus* β-glucosidases, subtle structural variations may account for its altered substrate specificity; further structural and mutational analyses are required to validate this hypothesis [34,51]. While beneficial for reducing contamination, operating at high temperatures necessitates specialized bioreactors and may increase energy costs. Fermentative biohydrogen scaling-up bioprocesses also requires careful monitoring and adherence to regulatory guidelines [55]. Based on the improvements seen over the years using genetic engineering, adaptive laboratory evolution, and fermentation process engineering, we anticipate that further optimization of the enzyme expression and pathway integration in *T. onnurineus* GH03 will lead to even higher H_2_ yields [37,56,57,58,59,60,61].

In this study, we introduced the *T. pacificus* GH gene cluster into *T. onnurineus* NA1, thereby constructing a GH03 mutant capable of effectively utilizing laminarin as a carbon source. The enhanced growth and hydrogen production observed in GH03 cultured in laminarin-based media were consistent with the previously reported exo-laminarinase activity of *Tpa*-GH [34]. While we observed elevated hydrogen production and laminarin utilization, the broader metabolic shifts in *T. onnurineus* GH03 remain to be clarified. Metabolomic or flux analysis could reveal how introducing the large GH cluster influences the central carbon metabolism, electron flow, and overall energy balance. However, our successful transformation of this long cluster suggests the feasibility of transferring large DNA fragments into hyperthermophilic archaea. Given the stability and broad substrate specificity of *Tpa*-GH, it is plausible that *T. onnurineus* GH03 could be further engineered, in conjunction with additional endo-acting glycoside hydrolases, to expand the range of marine polysaccharides targeted for saccharification. The strain would hold potential for biotechnological applications in high-temperature biomass conversion processes, paving the way for the more sustainable production of biofuels and value-added biochemicals.

## 4. Materials and Methods

### 4.1. Detection of Glycoside Hydrolase from Thermococcus Genomic Data

The archaeal genomic data of the *Thermococcus* species were collected from the NCBI database. After performing the protein-translated alignment, we selected 35 complete genomes, ensuring that only type strains were included. OrthoFinder version 2.4.0 [62] was used to obtain orthogroups from the *Thermococcus* genomes. The OrthoVenn3 v3.0 web server (https://orthovenn3.bioinfotoolkits.net/, accessed on 24 October 2024) was used for visualization and to perform orthologous analysis, examining orthologous clusters by comparing the genomes of two selected *Thermococcus* species: *T. onnurineus* NA1 and *T. pacificus* P-4 [63]. The OrthoFinder algorithm was also implemented in OrthoVenn3.

To identify and annotate orthogroups containing β-glucosyl hydrolase genes, each protein sequence within these orthogroups was then functionally annotated using InterProScan (https://www.ebi.ac.uk/interpro/search/sequence/ accessed on 11 November 2024) to establish protein families, domains, and functional sites. In addition, NCBI BLAST (https://blast.ncbi.nlm.nih.gov/Blast.cgi, accessed on 12 November 2024) was used and default parameters to validate the functional assignments. Orthogroups containing proteins annotated as β-glucosyl hydrolase were subsequently extracted, and the number of species including these orthogroups was quantified (Table 1).

### 4.2. Strains and Cell Culture Conditions

The wild-type strain *T. onnurineus* NA1 was cultivated anaerobically at 80 °C in nutrient-enriched ASW-YT medium [64], with additional organic compounds or elemental sulfur added when necessary. The ASW-YT medium comprised 0.8× artificial seawater, 5.0 g/L yeast extract, and 5.0 g/L tryptone. Resazurin (0.8 mg/L) was used as an indicator, and before inoculation, Na_2_S was added to reduce the medium until it became clear. For solid cultures, instead of elemental sulfur and Na_2_S-9H_2_O, 2 mL of polysulfide solution (prepared through dissolving 10 g of Na_2_S-9H_2_O and 3 g of sulfur in 15 mL of distilled water) per liter was used, and the medium was solidified using 10 g/L Gelrite.

For fosmid-based molecular cloning, *E. coli* EPI300™-T1R (Epicentre Biotechnologies, Madison, WI, USA) was employed. *E. coli* strains harboring fosmids were grown in LB medium supplemented with 12.5 μg/mL chloramphenicol.

### 4.3. Cloning and Mutant Construction

Standard molecular biology techniques and microbiological experiments following established methods were used [65]. The cloning strains and fosmids used in this study are listed in Appendix A. For the cloning of the large inserted gene cluster, we used the pNA1comFosC1096 fosmid vector backbone, which contained a complementary insertion site (LA (TON 1128-TON_1127) region-P_0157_ promotor-HMG cassette-RA (TON_1126) region), as described in a previous study [42]. The PCR products of the glycosyl hydrolase gene cluster in *T. pacificus* and fosmid vector pNA1comFosC1096 with *Avr*II enzyme digestion were assembled into a single vector using Gibson Assembly Master Mix (New England Biolabs, Ipswich, MA, USA). More specifically, the fusion targeting *Tpa*-GH region I and *Tpa*-GH region II from *T. pacificus* were fused using a homologous recombination event with 26 bp complementary PCR primers to construct 24 kb GH cluster inserted fosmid during the gene assembly reaction by the Gibson Assembly method. The sequences of the *Tpa*-GH gene cluster inserted into the genome were verified through DNA sequencing. *T. onnurineus* strains with the constructed fosmid were transformed, and transformants were confirmed using the modified CaCl_2_ method for *Thermococcus kodakaraensis* KOD1 [64]. Briefly, *T. onnurineus* NA1 was cultivated in ASW-YT liquid medium for 12 h; after this, cells in the late exponential phase were collected using centrifugation with a 3 mL culture (17,000× *g*; 5 min) and resuspended in 200 μL of transformation buffer (80 mM CaCl_2_ in 0.8× modified ASW, excluding KH_2_PO_4_). The cell suspension was maintained on ice for 30 min. Subsequently, 3 μg of plasmid DNA dissolved in Tris-EDTA buffer was introduced into the mixture, and this was followed by incubation on ice for 1 h. Heat shock treatment was performed at 80 °C for 45 s, and the cells were immediately placed back on ice for an additional 10 min. As a negative control, an equivalent volume of Tris-EDTA buffer, without DNA, was added to the cell suspension. The transformed cells were then transferred to 20 mL of ASW-YT liquid medium for recovery. Cells were cultured in a medium containing 10 μM simvastatin, which served as the selection marker. Following incubation, the culture was spread across ASW-YT agar plates and incubated at 80 °C for 48 h to allow for colony formation. The primers used for gene cloning are listed in Appendix A.

### 4.4. Growth and Hydrogen Production Assays

For the growth assays, *T. onnurineus* GH03 and its wild-type strain NA1 were first grown in YPS (yeast extract–peptone–sulfur) medium at 80 °C to obtain seed cultures, as previously reported [35]. These seed cultures were then transferred into a modified MM1 medium supplemented with 0.5 g/L yeast extract, along with either 0.5% (*w*/*v*) carboxymethylcellulose (MM1-CMC), 0.5% (*w*/*v*) agarose (MM1-Aga), or 0.25% (*w*/*v*) laminarin (MM1-Lam). The modified MM1 medium used in this study was prepared as described by Sokolova et al. (2004) and Kim et al. (2010) [66,67]. The composition of the medium was as follows: NaCl (35 g/L), KCl (0.7 g/L), MgSO_4_ (3.9 g/L), CaCl_2_·2H_2_O (0.4 g/L), NH_4_Cl (0.3 g/L), Na_2_HPO_4_ (0.15 g/L), Na_2_SiO₃ (0.03 g/L), NaHCO₃ (0.5 g/L), cysteine-HCl (0.5 g/L), Holden’s trace elements/Fe-EDTA solution (1 mL) [68], Balch’s vitamin solution (1 mL) [69].

Each subculture was incubated at 80 °C for 48 h and successively transferred twice under the same conditions. Following this, the cultures were moved again to adjusted MM1 medium containing 0.1 g/L yeast extract (with the same concentrations of CMC, agarose, or laminarin) and subjected to 2 more 48 h subculturing periods at 80 °C to assess the growth and degradation capabilities of both the mutant and wild-type strains.

Cell growth was quantified after each 48 h incubation period by measuring the optical density at 600 nm (OD_600_) with a UV–visible spectrophotometer (Biophotometer Plus; Eppendorf, Hamburg, Germany). Hydrogen production was evaluated through collecting 100 μL of headspace gas from each culture vessel and analyzing it using a gas chromatograph (YL 6100; YL Instrument Co., Anyang, Republic of Korea) equipped with Porapak N and Molecular Sieve 13X columns (Merck Korea, Seoul, Republic of Korea).

## Figures and Tables

**Figure 4 ijms-26-03305-f004:**
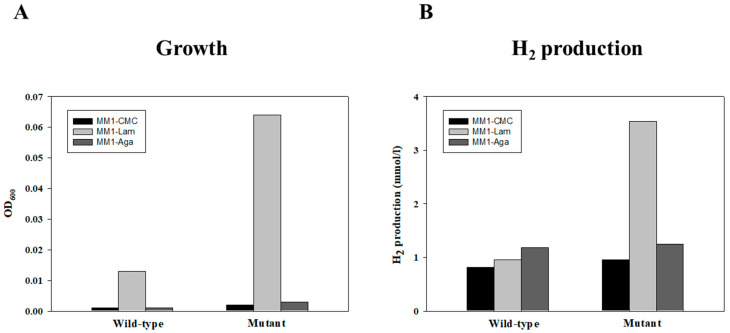
Growth and H_2_ production of wild-type and mutant GH03. (**A**) Growth (OD_600_) on MM1-CMC (carboxymethylcellulose), MM1-Lam (laminarin), and MM1-Aga (agarose). (**B**) H_2_ production (mmol/L) under the same culture conditions. Each measurement was performed in duplicate.

## Data Availability

Data contained within the article.

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
