# Peer review of "Ortholog Analysis and Transformation of Glycoside Hydrolase Genes in Hyperthermophilic Archaeal Thermococcus Species"

_ijms, 2025, doi:10.3390/ijms26073305_

Round 1

Reviewer 1 Report

Comments and Suggestions for Authors

Clarity, Relevance, and Structure

The manuscript is well-structured and addresses a relevant topic in the field of hyperthermophilic archaea and their applications in dihydrogen production. The study focuses on the transformation of glycoside hydrolase genes in Thermococcus species, which is significant for enhancing biomass conversion for dihydrogen production. The introduction provides a clear background, and the subsequent sections logically build.

References

The references cited are mostly recent, within the last five years for about 20%, within the last 15 years for about 80% and only one historical reference (1976) which is appropriate for a not so quickly evolving field. The citations are relevant to the study and support the arguments presented. There does not appear to be an excessive number of self-citations. The self-citations are mostly used not for rewriting technical points. The references are well-balanced with external sources.

Scientific Soundness and Experimental Design

The manuscript is scientifically sound, and the experimental design is appropriate for testing the two hypotheses. The study involves ortholog analysis, gene cloning, and transformation, followed by growth and dihydrogen production assays. These methods are suitable for investigating the integration and expression of glycoside hydrolase genes in Thermococcus species.

Reproducibility

The method section does not provide enough details to reproduce experiences without needing to search for several references (36.42). And this in particular for cloning and construction methods of mutants. More specific information on methods would allow the novice reader to save time on the study of this publication (the specialist reader already knows these publications!).

Figures, Tables, and Data Interpretation

The figures and tables are appropriate and effectively present the data. They are easy to interpret and understand, and the data is consistently interpreted throughout the manuscript. It is regrettable that in Figure 2 you use it and RA while in Figure 3 you use 1127LA and 1126ra. Homogenizing the terms between the two figures would facilitate reading.

Conclusions

The conclusions are consistent with the evidence and arguments presented. The study demonstrates the feasibility of stable, large-fragment DNA integration in hyperthermophilic archaea and the potential of T. onnurineus GH03 for high-temperature biomass conversion.

Summary

The manuscript is well-written, scientifically sound, and presents a relevant study in the field of hyperthermophilic archaea. The experimental design is appropriate, and the results are reproducible based on the methods provided. The conclusions are consistent with the evidence presented. The manuscript makes a valuable contribution to the field.

Author Response

Comments 1: [The method section does not provide enough details to reproduce experiences without needing to search for several references (36.42). And this in particular for cloning and construction methods of mutants. More specific information on methods would allow the novice reader to save time on the study of this publication (the specialist reader already knows these publications!).]

Response 1: [We agree that providing sufficiently detailed protocols is crucial for reproducibility, particularly for readers less familiar with these techniques. In response to your comment, we have substantially revised Section 4.3, ‘Cloning and Mutant Construction’.]

Comments 2: [The figures and tables are appropriate and effectively present the data. They are easy to interpret and understand, and the data is consistently interpreted throughout the manuscript. It is regrettable that in Figure 2 you use it and RA while in Figure 3 you use 1127LA and 1126ra. Homogenizing the terms between the two figures would facilitate reading.]

Response 2: [To ensure consistency and clarity, we have revised the labels in Figure 3 so that 1127LA is now referred to as LA and 1126ra as RA, matching the labeling convention used in Figure 2.]

Reviewer 2 Report

Comments and Suggestions for Authors

The study presents an important advancement in archaeal genetic engineering, demonstrating the stable integration of a large glycoside hydrolase (GH) gene cluster into Thermococcus onnurineus NA1. The findings highlight the enhanced ability of the engineered strain to degrade β-linked polysaccharides, leading to improved hydrogen production. However, several aspects could be clarified or expanded:

  1. In introduction second para, please add a line about the difference between thermophiles and hyper thermophiles.
  2. While the study successfully integrates the Tpa-GH gene cluster and demonstrates its impact on growth and hydrogen production, additional controls (e.g., a strain with an empty fosmid vector) would help confirm that the observed effects are specifically due to GH cluster expression rather than unintended genomic modifications.
  3. Page2, the authors wrote, “β-glucosidase is an important en-zyme that hydrolyzes various glycosidic bonds, converting them into glucose”. Please write different, the specific bond β-glucosidase cleave, and which bond authors are talking about in this manuscript for e. g.  beta-1,4 glycosidic bond.
  1. The long-term stability of the integrated GH cluster is not discussed. Addressing whether the inserted genes remain functionally expressed over multiple generations would be valuable for assessing potential industrial applications. Additionally, providing gene expression data (e.g., RT-qPCR or RNA-seq) would further support the functional impact of the introduced genes.
  2. While the study shows improved hydrogen production, comparing these results with the studies already done in other engineered or naturally occurring hydrogen-producing strains would provide better context for its industrial relevance. How does T. onnurineus GH03 compare with other known thermophiles in terms of efficiency?
  3. The discussion hypothesizes that the altered substrate specificity of Tpa-GH, compared to P. furiosus β-glucosidases, is due to structural differences in a loop region. While this is a reasonable explanation, additional structural  or modelling of the protein and showing a 3D structure to better explain the how different the loop region would be beneficial.
  4. The ability to transfer a >20 kbp gene cluster is a significant advancement. However, there is little mention of the stability of this cluster over multiple generations. Addressing whether the inserted genes remain functionally expressed over prolonged culture periods would be useful for assessing long-term applicability.
  5. The discussion contextualizes Thermococcus in marine environments and its role in carbon cycling. However, comparing its laminarin-degrading efficiency with other known thermophiles or engineered microbes would provide a better sense of its relative potential for industrial applications.
  6.   Future Applications and Limitations – While the study highlights the potential for expanding substrate specificity through further engineering, it would be helpful to briefly mention any technical challenges or limitations that may arise in applying this system at an industrial scale.
Comments on the Quality of English Language

NA

Author Response

Comments 1: [In introduction second para, please add a line about the difference between thermophiles and hyper thermophiles.]

Response 1: [In the second paragraph of the Introduction, we have added the following sentence to highlight their distinct growth temperature ranges and relevance to industrial applications:“Whereas thermophiles generally grow optimally between 45oC and 70oC, these hyperthermophiles exhibit an even greater thermal tolerance and typically thrive in extremely hot environments above 80oC” (Line 45-47)]

Comments 2: [While the study successfully integrates the Tpa-GH gene cluster and demonstrates its impact on growth and hydrogen production, additional controls (e.g., a strain with an empty fosmid vector) would help confirm that the observed effects are specifically due to GH cluster expression rather than unintended genomic modifications.]

Response 2: [We appreciate the suggestion for an additional control. As you know, in our experimental setup, the fosmid vector serves as a vehicle to introduce the target gene cluster into the host genome. Once the Tpa-GH gene cluster is integrated into the host genome, the remaining vector backbone is effectively lost and does not persist in the cell. As a result, in order to obtain a strain with an empty fosmid vector, it will be necessary to either develop an episomal vector (for NA1 strain) or create a recombinant plasmid for genome integration. Our integration strategy eliminates extraneous plasmid or vector elements, thus minimizing the potential unintended effects of the vector itself. This method ensures that the observed phenotypes—namely, enhanced growth on β-linked polysaccharides and improved hydrogen production—are attributable to the integrated Tpa-GH genes rather than vector-derived artifacts. Consequently, an empty fosmid control would neither remain stable nor provide additional clarity under these circumstances. We hope this clarification aligns with the rationale behind our design and addresses the reviewer’s concern regarding the potential off-target effects of the cloning vector.]

Comments 3: [Page2, the authors wrote, “β-glucosidase is an important enzyme that hydrolyzes various glycosidic bonds, converting them into glucose”. Please write different, the specific bond β-glucosidase cleave, and which bond authors are talking about in this manuscript for e. g.  beta-1,4 glycosidic bond.]

Response 3: [We have revised the statement on page 2 from “β-glucosidase is an important enzyme that hydrolyzes various glycosidic bonds, converting them into glucose” to “β-Glucosidase specifically hydrolyzes β-1,4 glycosidic bonds in substrates such as cellulose or laminarin, releasing glucose monomers.”]

Comments 4: [This change clarifies the precise bond cleavage relevant to our study and more accurately reflects the enzyme’s activity. The long-term stability of the integrated GH cluster is not discussed. Addressing whether the inserted genes remain functionally expressed over multiple generations would be valuable for assessing potential industrial applications. Additionally, providing gene expression data (e.g., RT-qPCR or RNA-seq) would further support the functional impact of the introduced genes.]

Response 4: [We have confidence in the stability of the integrated GH cluster from two perspectives. Firstly, the selection marker system (hmg cassette) employed in this study has been previously validated in several studies (NA1 or KOD1 strains, Yang et al., 2022; Hileman & Santangelo, 2012). Secondly, the observed improvements in growth performance and hydrogen production in laminarin-based media strongly indicate a functional impact of the introduced genes. We are currently conducting adaptation studies on the strain, and although these results have not yet been published, our ongoing work confirms that the integrated gene cluster remains stable over multiple generations. In this study, our focus was directed toward application-oriented outcomes rather than detailed gene expression analyses. We plan to include comprehensive gene expression data in future work to further support these findings.]

Comments 5: [While the study shows improved hydrogen production, comparing these results with the studies already done in other engineered or naturally occurring hydrogen-producing strains would provide better context for its industrial relevance. How does T. onnurineus GH03 compare with other known thermophiles in terms of efficiency?]

Response 5: [T. onnurineus NA1 is indeed well-recognized for its robust hydrogen production using formate as a substrate; however, it previously showed limited ability to utilize β-linked polysaccharides. Our work focuses on expanding this strain’s substrate range by engineering glycoside hydrolase activity. Consequently, compared to other naturally occurring thermophiles that are inherently adept at polysaccharide degradation, T. onnurineus GH03 may exhibit lower initial hydrolytic efficiency. Nonetheless, we believe the potential to combine NA1’s established hydrogen-production pathway with newly introduced glycoside-processing capabilities is highly promising. Ongoing optimization of enzyme expression and pathway integration will likely improve yields and strengthen T. onnurineus GH03’s competitiveness for industrial applications. We have added the following statement to the discussion section: "Based on the improvements seen over the years using genetic engineering, adaptive laboratory evolution, and fermentation process engineering, we anticipate that further optimization of enzyme expression and pathway integration in T. onnurineus GH03 will lead to even higher Hâ‚‚ yields" (Line 219-222)]

Comments 6: [The discussion hypothesizes that the altered substrate specificity of Tpa-GH, compared to P. furiosus β-glucosidases, is due to structural differences in a loop region. While this is a reasonable explanation, additional structural or modelling of the protein and showing a 3D structure to better explain the how different the loop region would be beneficial.]

Response 6: [We agree that the detailed discussion of structural differences between Tpa-GH and P. furiosus β-glucosidases was inadequately supported. In response to your comment, we have revised our Discussion by removing the extensive description of these structural details. Instead, we have added the following concise statement to the limitations section: "Although Tpa-GH exhibits high primary-sequence similarity to Pyrococcus furiosus β-glucosidases, subtle structural variations may account for its altered substrate specificity; further structural and mutational analyses are required to validate this hypothesis." (Line 214-216)]

Comments 7: [The ability to transfer a >20 kbp gene cluster is a significant advancement. However, there is little mention of the stability of this cluster over multiple generations. Addressing whether the inserted genes remain functionally expressed over prolonged culture periods would be useful for assessing long-term applicability.]

Response 7: [As mentioned earlier, we are currently conducting adaptation studies to evaluate the stability of the integrated gene cluster over multiple generations. Preliminary results indicate that the cluster remains stable during prolonged culture (data not shown), and we are in the process of finalizing these experiments for future reporting.]

Comments 8: [The discussion contextualizes Thermococcus in marine environments and its role in carbon cycling. However, comparing its laminarin-degrading efficiency with other known thermophiles or engineered microbes would provide a better sense of its relative potential for industrial applications.]

Response 8: [We acknowledge that Thermococcus currently does not exhibit as high a laminarin-degrading efficiency as some other known thermophiles or engineered strains. In our revised Discussion, we note that further adaptive evolution or metabolic engineering could be pursued to enhance T. onnurineus GH03’s laminarin hydrolysis capacity for industrial applications. We have added the following statement to the limitations section: “Moreover, its laminarin-degrading efficiency remained lower than that of strains specifically optimized for laminarin utilization, suggesting that further adaptive evolution or metabolic engineering could improve laminarin hydrolysis.” (Line 211-214)]

Comments 9: [Future Applications and Limitations – While the study highlights the potential for expanding substrate specificity through further engineering, it would be helpful to briefly mention any technical challenges or limitations that may arise in applying this system at an industrial scale.]

Response 9: [We have included the above sentences in our Discussion to emphasize both the need for specialized, high-temperature bioreactors—due to potential energy cost implications—and the regulatory considerations when deploying genetically modified hyperthermophiles on a large scale. This addition underscores the practical aspects that must be managed for successful industrial implementation. We have added the following statement to the limitations section: “While beneficial for reducing contamination, operating at high temperatures necessitates specialized bioreactors and may increase energy costs. Scaling up bioprocesses with genetically modified hyperthermophiles also requires careful monitoring and adherence to regulatory guidelines.” (Line 216-219)]

Round 2

Reviewer 2 Report

Comments and Suggestions for Authors

NA

Comments on the Quality of English Language

NA